# Optimized OTSU Segmentation Algorithm-Based Temperature Feature Extraction Method for Infrared Images of Electrical Equipment

**DOI:** 10.3390/s24041126

**Published:** 2024-02-08

**Authors:** Xueli Liu, Zhanlong Zhang, Yuefeng Hao, Hui Zhao, Yu Yang

**Affiliations:** School of Electrical Engineering, Chongqing University, Chongqing 400044, China; 20153560@cqu.edu.cn (X.L.); 20201101076g@stu.cqu.edu.cn (Y.H.); 20191101372@stu.cqu.edu.cn (H.Z.); 20163724@cqu.edu.cn (Y.Y.)

**Keywords:** power equipment, infrared image, segmentation, temperature feature extraction

## Abstract

Infrared image processing is an effective method for diagnosing faults in electrical equipment, in which target device segmentation and temperature feature extraction are key steps. Target device segmentation separates the device to be diagnosed from the image, while temperature feature extraction analyzes whether the device is overheating and has potential faults. However, the segmentation of infrared images of electrical equipment is slow due to issues such as high computational complexity, and the temperature information extracted lacks accuracy due to the insufficient consideration of the non-linear relationship between the image grayscale and temperature. Therefore, in this study, we propose an optimized maximum between-class variance thresholding method (OTSU) segmentation algorithm based on the Gray Wolf Optimization (GWO) algorithm, which accelerates the segmentation speed by optimizing the threshold determination process using OTSU. The experimental results show that compared to the non-optimized method, the optimized segmentation method increases the threshold calculation time by more than 83.99% while maintaining similar segmentation results. Based on this, to address the issue of insufficient accuracy in temperature feature extraction, we propose a temperature value extraction method for infrared images based on the K-nearest neighbor (KNN) algorithm. The experimental results demonstrate that compared to traditional linear methods, this method achieves a 73.68% improvement in the maximum residual absolute value of the extracted temperature values and a 78.95% improvement in the average residual absolute value.

## 1. Introduction

Due to its advantages of visualization, convenience, high sensitivity, and non-contact temperature measurement, infrared thermography technology has been widely applied in monitoring the operating status of, and detecting faults in, power system equipment [1]. Real-time and accurate infrared image monitoring of electrical equipment is crucial for ensuring stable operation [2,3,4]. Improving real-time capability allows for the timely detection of equipment faults or potential issues, while enhancing accuracy reduces false alarms and prevents wasting the time and energy of maintenance personnel. However, with the increasing number of power grid devices and maintenance requirements, traditional manual image recognition methods for infrared images are no longer sufficient to meet the demands of large-scale fault diagnosis of power equipment [5]. Moreover, manual image recognition results are influenced by factors such as inspection personnels’ experience, expertise, and fatigue, which can lead to a misdiagnosis. With the rapid development of computer technology and artificial intelligence, utilizing AI technology to extract temperature features from massive infrared images for monitoring, analysis, and intelligent diagnosis has become a trend and holds significant research value and application prospects. However, the key challenge lies in how to extract temperature features quickly and accurately, and then accurately represent the equipment’s status.

Extracting temperature features of electrical equipment from infrared images involves two steps: device segmentation and temperature feature extraction. Among the various image segmentation methods, extensively utilized threshold-based, region-based, and edge-based segmentation can be found [6,7,8,9]. Among numerous image segmentation methods, the maximum between-class variance thresholding method, known as the OTSU method [10,11,12,13], proposed by Nobuyuki OTSU from Japan, is regarded as the optimal algorithm for selecting thresholds in image segmentation due to its simplicity, high accuracy, and independence from image brightness and contrast. As a result, it has gained significant popularity in the field of digital image processing. However, calculating the optimal threshold by iterating over pixel values requires a significant amount of computation, reducing the efficiency of image segmentation. In order to accelerate the segmentation process, scholars from both domestic and international communities have proposed various intelligent optimization algorithms, such as particle swarm optimization (PSO) [14], Grey Wolf Optimization (GWO) [15], and the genetic algorithm (GA) [16], which have lower computational costs. Huang et al. improved the OTSU method by using the fruit fly optimization algorithm, achieving good results [17]. Ning introduced the whale optimization algorithm into threshold image segmentation, speeding up the segmentation process and achieving satisfactory results [18].

After completing the image segmentation, temperature information can be extracted from the segmented regions. The matrix data obtained using infrared instruments contain the temperature information of the object, which is converted from the temperature matrix to the RGB image via a pseudo color transformation. The temperature information in infrared images is usually extracted using the accompanying infrared image analysis software, but this software is usually expensive and lacks universality. The color bar is an important medium for converting temperature matrices to infrared images, so the temperature information of each point in the image can be obtained from the color bar in the infrared image [19]. Traditional infrared image temperature calculation approximates the grayscale and temperature values of each pixel on the image as a linear function. By fitting the obtained linear function relationship, the temperature value for each pixel is determined. Zheng et al. studied the function relationship between pixel grayscale and temperature for the FLIR T640 infrared thermal imager, using grayscale values as independent variables and temperature values as dependent variables to fit a linear function curve for the temperature extraction of power equipment infrared images [19]. However, the grayscale and temperature of the infrared image do not have a strictly linear correspondence, so the accuracy of temperature estimation using linear functions still needs improvement.

In summary, the extraction of temperature features of target devices from infrared images mainly involves image segmentation and temperature extraction. When using the OTSU method as a key component in image segmentation, the computational complexity is high and the segmentation speed needs improvement. When extracting temperature information using the relationship between grayscale and temperature, there is limited research on their non-linear relationship and the extraction accuracy needs improvement. To address these two issues, in this study, we propose a temperature feature extraction method for infrared images of electrical equipment based on an optimized OTSU algorithm. First, the Gray Wolf Optimization (GWO) algorithm is used to optimize the threshold determination process within the traditional OTSU segmentation method, resulting in an improved OTSU segmentation algorithm based on GWO. This enhances the segmentation speed and separates the target device regions in the infrared image. Based on this, we propose a temperature extraction method for infrared images using the K-nearest neighbor (KNN) algorithm to improve the temperature value extraction accuracy, obtaining a temperature value feature vector that includes the highest temperature, lowest temperature, and average temperature of the device on the image. The proposed method provides a reference for real-time and accurate infrared monitoring of electrical equipment. The workflow of this method is shown in Figure 1.

## 2. The Infrared Image Segmentation Algorithm

In order to improve the segmentation speed, the GWO optimization algorithm was utilized to optimize the threshold determination process in OTSU. The optimized threshold obtained through this process was then used for infrared image segmentation.

### 2.1. OTSU Algorithm Principle

The OTSU algorithm [20], also known as the maximum inter-class variance method, was proposed by Otsu in 1979. It is a widely acclaimed algorithm for threshold selection in image segmentation due to its simplicity and robustness, making it highly popular in digital image processing.

Assuming the size of the image is *M* × *N*, the optimal threshold for a binary conversion is T, which divides the image into two categories: background and target. The number of pixels belonging to the background is *N*_1_, the number of pixels belonging to the target is *N*_2_, the ratio of the background pixels in the entire image is *ω*_1_, the grayscale value of the background is *μ*_1_, the ratio of the target pixels in the entire image is *ω*_2_, the grayscale value of the target is *μ*_2_, and the average grayscale value of the entire image is *μ*. Therefore [21]:(1)$$\omega_{1}=\frac{N_{1}}{M\cdot N}$$
(2)$$\omega_{2}=\frac{N_{2}}{M\cdot N}$$
(3)$$N_{1}+N_{2}=M\cdot N$$

Employing Equations (1)–(3), we obtain:(4)$$\omega_{1}+\omega_{2}=1$$
(5)$$\begin{array}{c} \mu =\frac{\mu_{1}\cdot N_{1}+\mu_{2}\cdot N_{2}}{M\cdot N}\\ \mu_{1}\omega_{1}+\mu_{2}\omega_{2} \end{array}$$

The formula for between-class variance is as follows:(6)$$\sigma_{B}^{2}=\omega_{1}{\left(\mu_{1}-\mu_{T}\right)}^{2}+\omega_{2}{\left(\mu_{2}-\mu_{T}\right)}^{2}$$

It can be equivalent to:(7)$$\sigma_{B}^{2}=\omega_{1}\omega_{2}{\left(\mu_{1}-\mu_{2}\right)}^{2}$$

Ideally, within the same class, the intra-class variance should be minimal, while the variance between the background and the target across classes should be maximal, indicating a significant distinction between the two components comprising the image. Consequently, the threshold value that maximizes the variance between the background and the target is determined by iteratively exploring various threshold values, leading to the desired outcome.

### 2.2. The GWO-Optimized OTSU Segmentation Algorithm

GWO is a novel swarm intelligence algorithm proposed by Australian researchers based on their observation of the hunting behavior and hierarchical structure of wolf packs in the natural world [15].

In the GWO algorithm, the wolf pack is divided into four hierarchical levels in a pyramid shape, as shown in Figure 2. The leaders of the pack, represented by levels α, β and γ, have a more acute perception of potential prey locations compared to other wolves. They lead the pack in searching, tracking, and approaching the prey.

During the process of optimizing parameters using the GWO algorithm, the positions of the current time for levels α, β and γ are defined as the three best solutions found so far, while the position of the prey represents the actual best solution within the search range. The optimization process of the GWO algorithm is essentially a process of searching for the best solution within the search range based on the current best solutions [15].

The advantages of GWO lie in its simplicity and efficiency. It does not require complex parameter settings and has a fast convergence speed. Additionally, GWO exhibits good global search capability and convergence performance, making it capable of achieving good results for various optimization problems.

The traditional OTSU algorithm involves sequentially traversing all pixel values in an image to obtain the optimal threshold, which can be time-consuming. To speed up the segmentation process, the GWO optimization algorithm is employed to optimize the process of finding the optimal threshold. OTSU’s inter-class variance function is used as the fitness function, with population individuals representing pixel values. By iteratively updating the positions of the initial population, a new population is obtained. Each iteration produces a population with better fitness values than the previous generation, and after reaching the maximum number of iterations, the population represents the optimal threshold.

Utilizing the GWO algorithm, the optimal threshold is determined, enabling the binarization of the image. This process involves assigning a value of 1 to pixels with grayscale values surpassing the optimal threshold, while pixels with grayscale values lower than the optimal threshold are set to 0. As a result, the image is segmented, yielding a binary representation. The pseudocode of GWO algorithm is shown in Algorithm 1 and the flowchart of the GWO-optimized OTSU segmentation algorithm is shown in Figure 3.
**Algorithm 1: GWO Algorithm Pseudocode**1. Input:   population_size—size of the population;   num_iterations—number of iterations;   lower_bound—lower bound for the variables;   upper_bound—upper bound for the variables.2. Initialization:Create a population of size population_size and randomly initialize the position and fitness value for each individual;   Compute the fitness value for each individual.3. Find the optimal solution:For each iteration, do the following:For each individual, compute the fitness value;Find the best individual with the highest fitness value in the current population, denoted as alpha;Find the second best individual with the second highest fitness value in the current population, denoted as beta;Find the worst individual with the lowest fitness value in the current population, denoted as delta;For each individual in the population, update the position based on the position of alpha, beta, and delta:For each dimension, compute the new position;If the new position is out of bounds, set it to the boundary value;   Return the position of the best individual with the highest fitness value.4. Main program:Initialize the population;For each iteration, do the following:Find the position of the best individual;Output the current iteration number and the fitness value of the best solution;Update the population.5. Output:   The position and fitness value of the best solution.


## 3. Infrared Image Temperature Feature Extraction Method Based on an Optimized OTSU Algorithm

Extracting the temperature information of the segmented regions allows us to obtain the temperature features of the target device. In infrared images, the temperature data corresponding to each pixel are stored in matrix form. The color bar serves as a temperature reference in the image, where each temperature point on the color bar theoretically corresponds to a temperature point in the image. By associating the temperature parameters with the image grayscale using the color bar, we can obtain the temperature value of each point in the image. In this section, the KNN algorithm is employed to extract the temperature values of the pixels, thereby obtaining the temperature feature vector of the target device.

### 3.1. Traditional Linear Temperature Extraction Method

After converting the infrared image into grayscale, the temperature information of the device is reflected in the grayscale values. There exists a certain monotonic relationship between the temperature values and the grayscale values. The formula for converting the RGB three-channel infrared image to grayscale is as follows:(8)Gray=0.299R+0.587G+0.114B
where *Gray* is the pixel value after grayscale conversion, and *R*, *G*, *B* are the pixel values of the three channels of the infrared image.

By leveraging the temperature range provided by the color bar, comprising the maximum and minimum values, it is possible to establish the correlation between the temperature values and the grayscale values.

The function relationship between the temperature values and the grayscale values is:(9)$$\begin{array}{l} T=kGray+b\\ k=\frac{t_{\max}-t_{\min}}{G_{\max}-G_{\min}}\\ b=t_{\max}-\frac{t_{\max}-t_{\min}}{G_{\max}-G_{\min}}\cdot G_{\max} \end{array}$$

In the formula, *T* represents the temperature value, *Gray* represents the grayscale value, *t_max_* and *t_min_* represent the upper and lower limits of the temperature value on the color bar, and *G_max_* and *G_min_* represent the maximum and minimum grayscale values on the color bar, respectively.

### 3.2. KNN-Based Infrared Image Temperature Value Extraction Method

Traditional linear methods for extracting temperature values from infrared images calculate the temperature of each point on the image based on a linear relationship between the grayscale value and the temperature. However, since the grayscale value and temperature on an infrared image do not strictly follow a linear relationship, the accuracy of temperature values obtained using this method requires improvement. In this section, we propose a KNN-based method for extracting temperature values from infrared images.

KNN (K-nearest neighbor) is an algorithm introduced by Cover and Hart in 1968. The term “K-nearest neighbor” implies that each sample can be represented by its K-closest neighbor in the dataset [22]. As shown in Figure 4, when drawing a circle centered on a sample, since the highest number of shapes inside the circle are triangles, then the sample is considered a triangle.

The KNN algorithm usually uses the Euclidean distance as the distance metric. For two n-dimensional vectors in space, A(x_11_,x_12_,…,x_1n_) and B(x_21_,x_22_,…,x_2n_), the Euclidean distance between them is calculated as follows:(10)$$d_{AB}=\sqrt{\sum_{k=1}^{n}{\left(x_{1k}-x_{2k}\right)}^{2}}$$

In establishing the training set, the training data and their corresponding class labels must be determined. Then, the test data to be predicted are compared to the training set data one by one based on their features. The K-nearest data points are selected from the training set, and the classification with the most votes among these K data points is taken as the class of the new sample.

The temperature value of a point on the color bar is obtained using the highest and lowest temperatures on the color bar and the height of the color bar, according to the following formula:(11)$$t=(y-y_{min})t_{dis}+t_{min}$$
where *y_min_* is the y-coordinate of the lowest temperature point and *t* is the temperature value of point (*x*, *y*).

The gray value of point (*x*, *y*) is extracted, and a temperature value corresponding to the gray value is obtained. Following this method, a coordinate x is selected, and every pixel on the color bar is traversed from bottom to top to obtain all gray values and their corresponding temperature values, which are saved as a csv file.

By using the data in the csv file as the training set for the KNN algorithm, setting the K value to 1, and using the gray value of the measurement point as the algorithm input, the temperature of the measurement point can be obtained as the output. The flowchart of the KNN-based infrared image temperature extraction method proposed in this section is shown in Figure 5.

### 3.3. A Temperature Feature Extraction Method for Infrared Images Based on an Optimized OTSU Algorithm

This section proposes a temperature feature extraction method for infrared images based on an optimized OTSU algorithm. Following the segmentation of the infrared image using the method described in Section 2.2, the temperature extraction method from Section 3.2 is then applied to extract the three temperature features of the image: maximum temperature, minimum temperature, and average temperature.

Extraction of maximum and minimum temperatures

The process of extracting the maximum and minimum temperatures of the infrared image on the surface of the equipment is as follows: determine the region of the equipment, find the pixel with the maximum and minimum values, and extract the corresponding temperature values as the maximum and minimum temperatures.

2.Extraction of average temperature

Since the maximum and minimum temperatures can only reflect the local temperature of the power equipment and cannot fully reflect the state of the power equipment, the average temperature is considered another effective feature quantity of the equipment. By extracting the average temperature and combining it with the maximum temperature, minimum temperature, and maximum temperature rise, it becomes an effective temperature feature quantity of the power equipment. The process of extracting the average temperature is as follows: determine the area of the power equipment, extract all pixel values, accumulate the number of pixel points, calculate the average pixel value, determine the temperature value corresponding to the average pixel value, and obtain the average temperature.

## 4. Method Validation

In order to validate the viability of the proposed method in this article, infrared images of equipment such as insulators, transformers, and casings were captured using the FLIR T865 infrared thermal imager. The experimental environment consisted of Python 3.8, Windows 10, Intel(R) Core(TM) i5-7400 CPU @ 3.00 GHz CPU, and 8 GB RAM. The experimental setup had a population size of 80, a maximum iteration count of 50, and a search range of (0, 255).

For validation and analysis, an infrared image with a temperature range from 28.4 ± 1 °C to 9.2 ± 1 °C and a pixel resolution of 640 × 480 was selected as an example, as shown in Figure 6.

### 4.1. Infrared Image Segmentation

The algorithm’s performance was assessed by comparing both the runtime and the quality of image segmentation. To conduct an objective and scientific evaluation of the segmentation results, two widely used image quality evaluation metrics, peak signal-to-noise ratio (PSNR) and structural similarity (SSIM), were chosen.

In this study, infrared images of operating electrical equipment (Figure 6), and cropped portions of insulators, transformers, and casings from these infrared images were selected as experimental images. The four experimental images include an overall infrared image and localized infrared images. The original images of the four experimental images and their grayscale histograms are shown in Figure 7. From Figure 7, it can be seen that the histograms of infrared images have obvious double peaks, so the image can be segmented by setting a threshold. However, only the threshold range can be obtained from the histogram, and an accurate threshold cannot be obtained. Therefore, it is necessary to use the OTSU method to calculate an accurate optimal threshold.

The proposed GWO-OTSU algorithm was compared to the traditional OTSU algorithm, as well as the classical genetic algorithm, the sparrow optimization algorithm, and the whale optimization algorithm. Each algorithm was employed to derive the optimal threshold for the aforementioned experimental images, where by σ_B_^2^ is taken as the output of the fitness function for all optimization methods and the grayscale value that maximizes the output value, which is the optimal threshold, is identified. However, under identical conditions, the runtime of the same algorithm, represented as “t”, is not fixed but exhibits fluctuations. Therefore, to provide an objective evaluation, the same algorithm was run 100 times under the same hardware configuration, and the average runtime was calculated. The experimental results are shown in Table 1. Based on the optimal thresholds obtained, the images were segmented, and the segmentation results are shown in Figure 8.

As can be seen from Figure 8 and Algorithm 1, the binarization thresholds for the four experimental images are 90, 93, 112, and 96, respectively. Under the same evaluation criteria for PSNR and SSIM, the proposed GWO-OTSU algorithm reduced the average computation time for the optimal threshold by 83.99% compared to the traditional OTSU algorithm, while maintaining similar segmentation results. Although the other three optimization algorithms can also accurately determine the optimal threshold, their average runtime improvement rates were 68.07%, 70.32%, and 71.93%, respectively, which is more than 12% lower than the improvement rate of the proposed algorithm, indicating that they are less real-time than the proposed algorithm. Therefore, without sacrificing segmentation accuracy, the proposed algorithm exhibits a lower runtime compared to the traditional OTSU algorithm, enabling faster determination of the optimal threshold for image segmentation.

In order to discuss the statistical differences between the methods and further demonstrate the superiority of our method, we conducted experiments on 20 different infrared images of power equipment using the same method. These 20 infrared images of power equipment are from Appendix J of the DL/T 664-2016 Infrared Diagnosis Application Specification for Live Equipment [23], including bushings, transformers, capacitors, circuit breakers, lightning arresters, insulators, cables, clamps, isolating switches, and other equipment. The experimental image threshold, running time, and other results are shown in Table 2, and the binary image is shown in Figure 9.

From Table 2 and Figure 9, it can be seen that under the same segmentation effect, the proposed GWO-OTSU algorithm reduces the average calculation time of the optimal threshold by 84.10% compared to the traditional OTSU algorithm. The average runtime improvement rates of the other three optimization algorithms are 67.77%, 71.37%, and 71.85%, respectively, which is more than 12% lower than the improvement rate of the algorithm proposed in this article. Furthermore, their real-time performance is not as good as the method proposed in this article. Therefore, without sacrificing segmentation accuracy, compared to traditional OTSU algorithms, the proposed algorithm has a lower runtime and can quickly determine the optimal threshold for image segmentation.

### 4.2. Temperature Feature Vector Extraction

#### 4.2.1. Extraction of Temperature Values from Normal Infrared Images

The grayscale result of converting Figure 6 to grayscale is shown in Figure 10.

The temperature points of the color bar correspond one-to-one with the temperature points of the infrared image in theory, so analyzing the accuracy of temperature extraction on the color bar can help obtain the accuracy of temperature extraction of infrared images. Uniformly select 20 points on the color bar as sampling points for analysis.

The calculation method for the actual temperature of the sampling points in this article was as follows: Since the color bar is temperature-linear, the temperature of each point on the color bar should increase uniformly with the increase in the vertical coordinate. If the vertical coordinates of the starting and ending temperatures are determined, the temperature values of each intermediate point can be calculated. In Figure 9, if the coordinates of the upper left corner are set to (0,0), then the coordinates of the lower right corner are (639,479). Within the range where the color bar is located, the vertical coordinates of the sudden change in grayscale value are taken as the vertical coordinates of the starting and ending temperatures. Therefore, the vertical coordinates corresponding to 28.4 °C are 30, and the vertical axis corresponding to 9.2 °C is 423. Therefore, for the color bar in Figure 9, starting from 30, the vertical coordinates increase by 1, and the temperature value decreases by 0.049 °C. In this way, the temperature values corresponding to each ordinate in the 30–423 range can be obtained, and 20 temperature values can be taken evenly from the middle as the actual temperature values of the sampling points.

The traditional linear and proposed methods are used to extract the temperature and grayscale relationship of the infrared image color bar, and the measured temperature values of each sampling point on the color bar are obtained. Then, error analysis is performed on the actual temperature value of the sampling point and the measured temperature value, and the analysis results are shown in Figure 11.

From the figure, it is evident that the measurement results obtained using the proposed method closely align with the actual values. The absolute residuals of the test data are shown in Figure 12.

From Figure 12, it can be observed that the absolute residuals of the proposed method are generally smaller compared to those of the traditional linear method. The maximum absolute residual for the traditional linear method is 0.57 °C, while the proposed method is only 0.15 °C, resulting in an improvement rate of 73.68%. The average residual for the traditional linear method is 0.19 °C, whereas the proposed method proposed is only 0.04 °C, resulting in an improvement rate of 78.95%. Hence, the proposed method exhibits higher accuracy compared to the traditional linear method.

A randomly selected rectangular region on the captured infrared image, with the coordinates of the top left and bottom right points being (288, 167) and (315, 186), respectively. Following the method described in Section 2.1, the maximum temperature value, minimum temperature value, and average temperature value of all points within the region are calculated as 28.35 °C, 26.69 °C, and 27.76 °C, respectively. The actual maximum, minimum, and average temperature values of all points within the rectangular box can be obtained from the accompanying infrared image analysis software, which are 28.383 °C, 26.652 °C, and 27.826 °C, respectively. It can be observed that the absolute residuals between the extracted minimum, maximum, and average temperature values and their actual values are 0.033 °C, 0.038 °C, and 0.066 °C, respectively. This indicates that the proposed method can accurately extract temperature features from infrared images. Therefore, the proposed method can be used for temperature extraction in infrared images and obtain high-precision temperature values.

In order to further demonstrate the effectiveness of the method proposed in this article, we conducted experiments on 20 infrared images of power equipment using the same method. These 20 infrared images of power equipment were captured using a FLIR infrared thermal imager, including bushings, transformer boxes, radiators, and oil conservators. The experimental image is shown in Figure 13, and the experimental residual absolute value analysis is shown in Figure 14.

From Figure 14, it can be seen that compared to traditional linear methods, the absolute residual values are generally smaller. In order to observe the errors of the two methods more intuitively, we calculated the experimental average absolute error and maximum residual absolute values for each image, as shown in Figure 15.

From Figure 15, it can be seen that compared to traditional linear methods, the proposed method has a smaller average absolute error and smaller maximum residual absolute values. Therefore, the proposed method can achieve higher accuracy in extracting temperature values from infrared images.

#### 4.2.2. Extraction of Temperature Values from Infrared Images with Added Noise

Due to the harsh operating environments of most power equipment, there may be various interfering factors during actual maintenance, leading to the presence of significant noise in the captured infrared images. This noise is primarily Gaussian noise, with a small amount of salt-and-pepper noise. In order to validate the effectiveness of the proposed temperature extraction method under the influence of noise, Gaussian noise with a mean of 0 and a variance of 2, as well as salt-and-pepper noise with a density of 0.002, were added to the captured infrared images. Figure 16 shows the images with the added noise.

Both the traditional linear method and the proposed method were used to extract the temperature and grayscale relationship from the color bar in the infrared images with added noise. The temperature measurement values for various points on the color bar were obtained. Error analysis was conducted on 20 uniformly selected points on the color bar, and the results are shown in Figure 17 and Figure 18.

From the above two figures, it can be observed that even in the presence of noise, the absolute residuals of the proposed method are still generally smaller than those of the traditional linear method.

Figure 19 shows a comparison of the temperature value fluctuations for each measurement point extracted using the proposed method and the traditional linear method before and after the addition of noise.

From Figure 19, it can be observed that before and after the addition of noise, the temperature values extracted utilizing the proposed method show minimal fluctuations for each measurement point. In contrast, the temperature values extracted employing the traditional linear method exhibit larger fluctuations. This indicates that the proposed method provides more stable test results and demonstrates better resistance to interference.

#### 4.2.3. Extraction of Temperature Feature Vectors from Power Equipment Infrared Images

A temperature feature vector T = [t_1_, t_2_, t_3_] was created, where t_1_ represents the maximum temperature, t_2_ represents the minimum temperature, and t_3_ represents the average temperature. The relevant temperature values of the infrared images of insulators, transformers, and bushings in Figure 6 were extracted according to the method outlined in Section 3.3 to realize the temperature feature extraction of the infrared image of the power equipment. The temperature feature vectors T_1_, T_2_, and T_3_ for the three power equipment infrared images are as follows:$$\begin{array}{c} T_{1}=[21.06\;{}^{\circ}C,13.90\;{}^{\circ}C,19.26\;{}^{\circ}C]\\ T_{2}=[28.40\;{}^{\circ}C,11.15\;{}^{\circ}C,23.59\;{}^{\circ}C]\\ T_{3}=[23.22\;{}^{\circ}C,14.96\;{}^{\circ}C,21.06\;{}^{\circ}C] \end{array}$$

## 5. Conclusions

In this study, we investigated real-time and accuracy issues during temperature feature extraction from power equipment infrared images, drawing the following two conclusions:By utilizing the Gray Wolf Optimization (GWO) algorithm to calculate the maximum inter-class variance threshold for the OTSU method, an optimized OTSU segmentation algorithm based on GWO is obtained. This algorithm improves the rate of finding the optimal segmentation threshold. The experimental results show that the proposed method reduces the average computation time for the optimal threshold by 83.99%, while maintaining a similar segmentation effect.By combining the K-nearest neighbor (KNN) algorithm, the temperature values from power equipment infrared images are extracted, addressing the issue of high errors in temperature calculation using traditional linear fitting methods. The experimental results show that compared to the traditional linear method, the proposed method achieves a 73.68% improvement in the absolute residuals and a 78.95% improvement in the average residuals. The proposed method, therefore, demonstrates higher accuracy compared to the traditional linear method.

This method may provide a reference for extracting temperature features from images in power equipment fault diagnosis. In future research, we can extract the temperature information of power equipment from infrared images and combine it with relevant industry and national standards to conduct fault prediction for power equipment. Furthermore, the classification of fault levels can be based on the temperature variations in power equipment captured in infrared images, thereby reducing human and material losses caused by equipment failures.

## Figures and Tables

**Figure 1 sensors-24-01126-f001:**
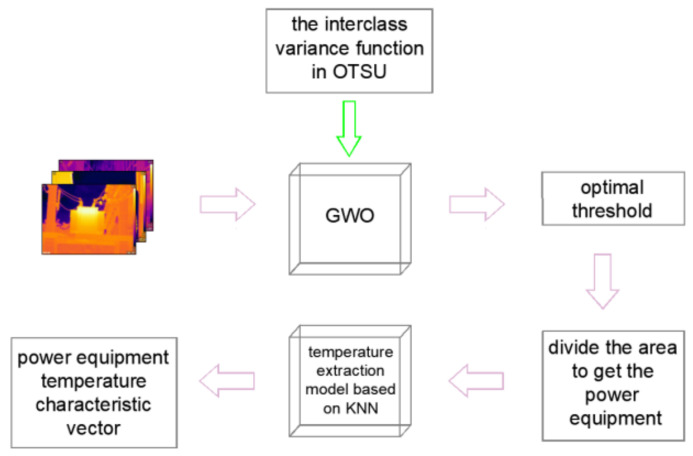
Flowchart for temperature feature extraction from power equipment infrared images.

**Figure 2 sensors-24-01126-f002:**
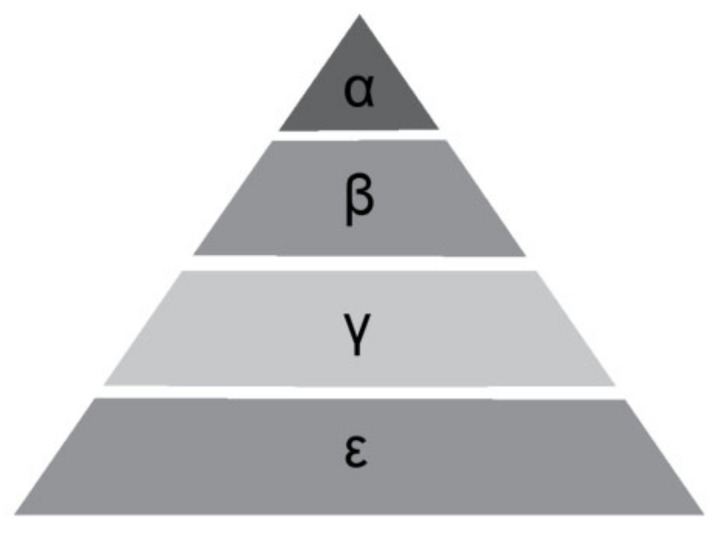
The Gray Wolf ranking system.

**Figure 3 sensors-24-01126-f003:**
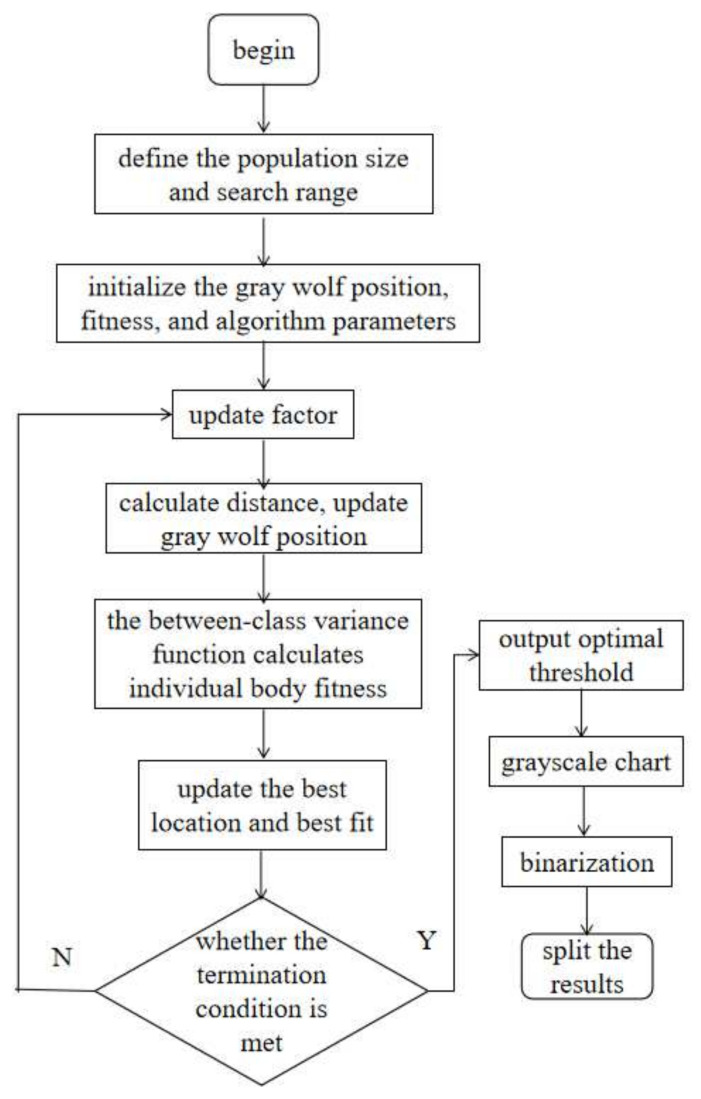
Flowchart of the GWO-optimized OTSU segmentation algorithm.

**Figure 4 sensors-24-01126-f004:**
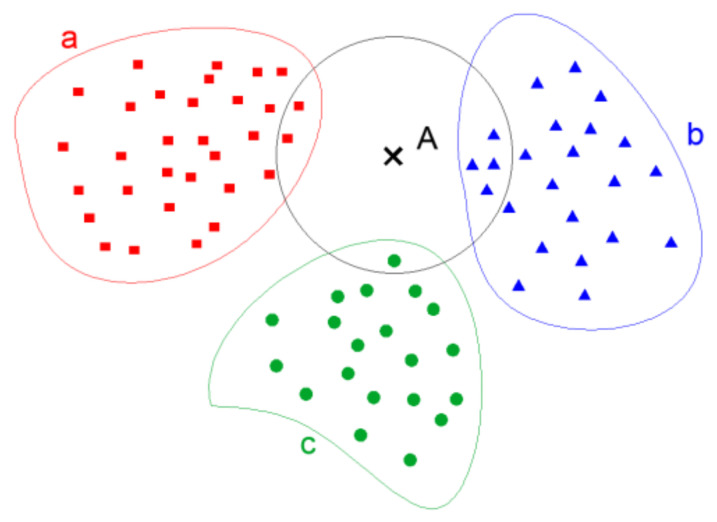
KNN classification diagram. a. b, c represent samples of known categories, A represents samples to be classified.

**Figure 5 sensors-24-01126-f005:**
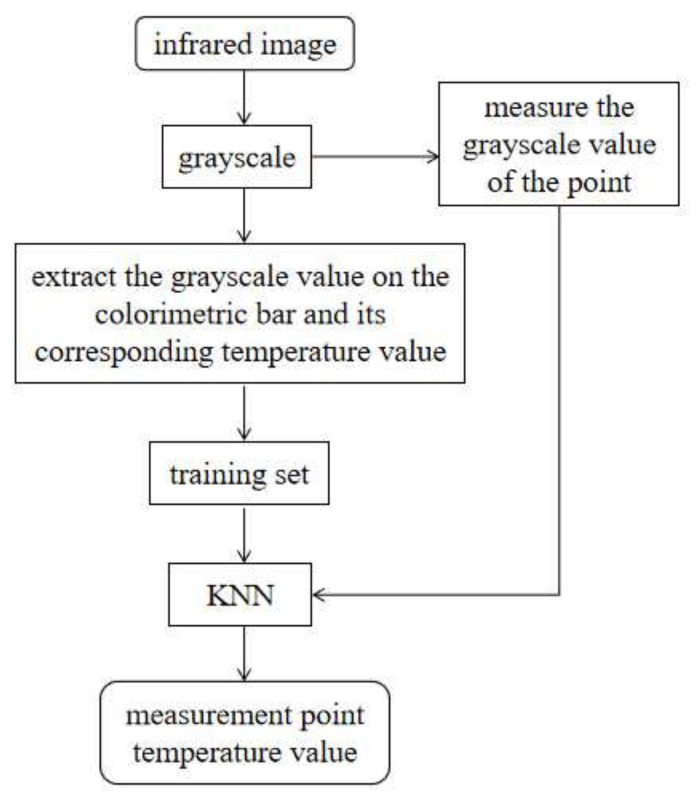
Flow diagram of infrared image temperature extraction method based on KNN.

**Figure 6 sensors-24-01126-f006:**
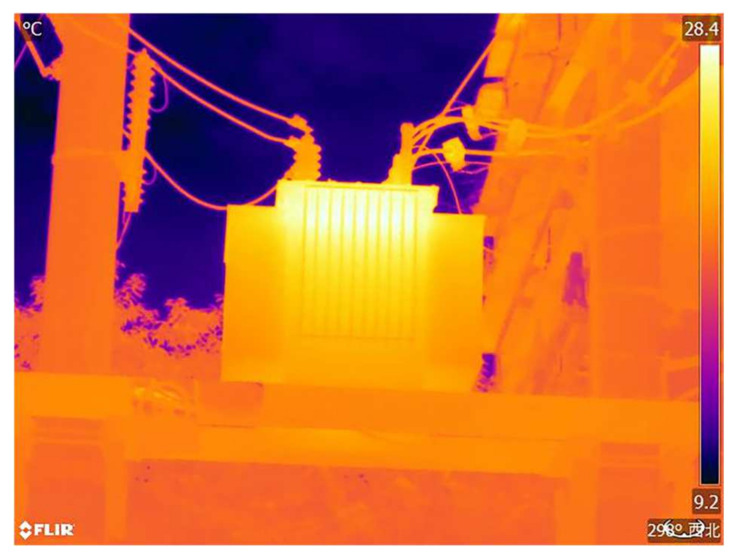
Power equipment in operation.

**Figure 7 sensors-24-01126-f007:**
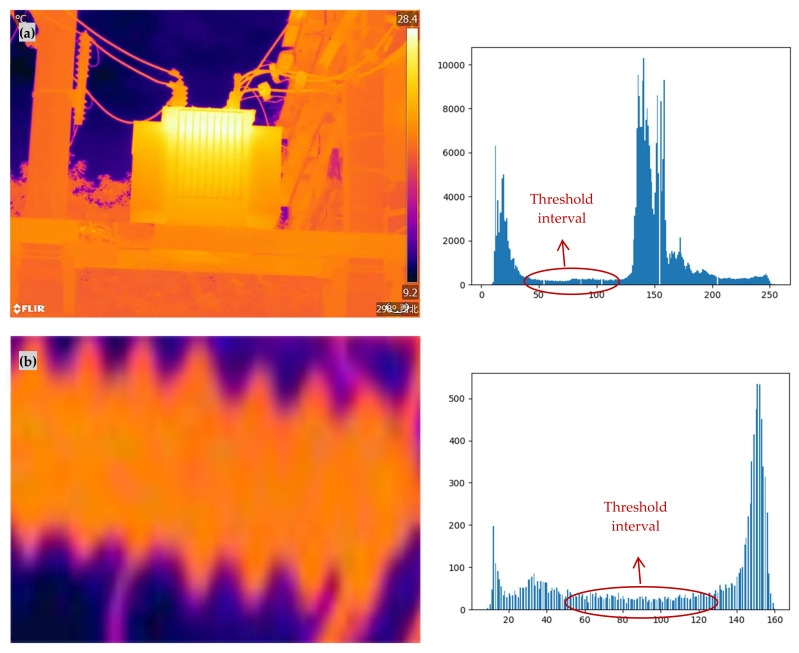
Experimental image and histogram thereof. (**a**) Infrared image of operational power equipment; (**b**) insulator intercepted in the equipment; (**c**) transformer intercepted in the equipment; (**d**) sleeve intercepted in the equipment.

**Figure 8 sensors-24-01126-f008:**
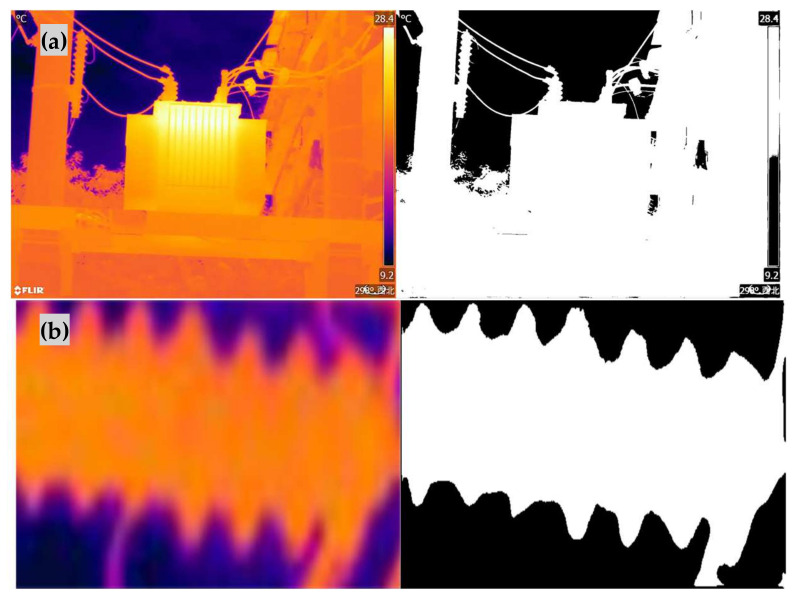
Experimental images and segmentation renderings. (**a**) Infrared image of operational power equipment; (**b**) insulator intercepted in the equipment; (**c**) transformer intercepted in the equipment; (**d**) sleeve intercepted in the equipment.

**Figure 9 sensors-24-01126-f009:**
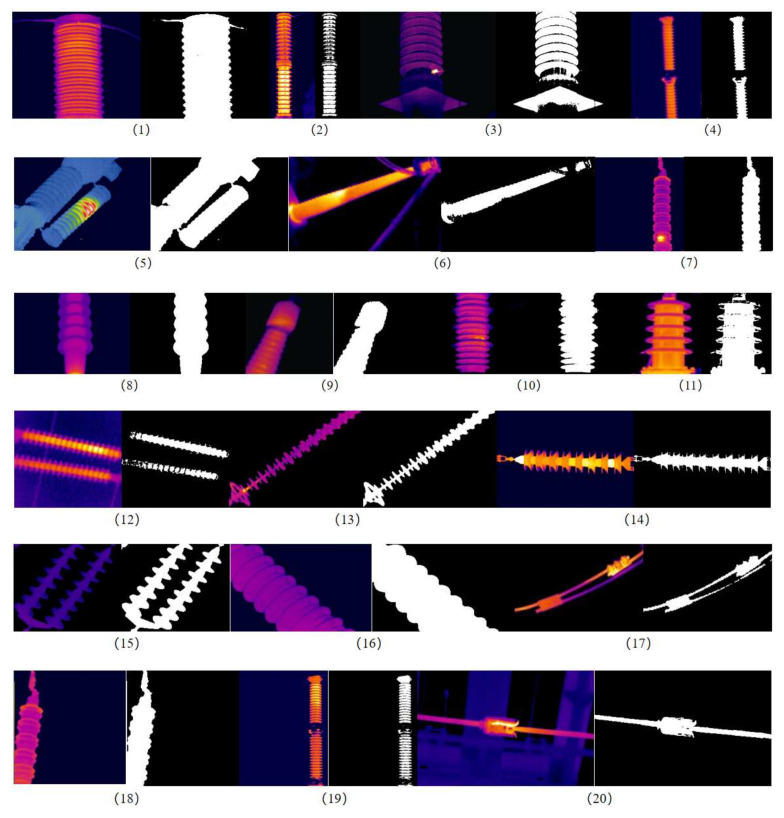
Infrared image and its segmented image. (1)–(20) represents the experimental image number.

**Figure 10 sensors-24-01126-f010:**
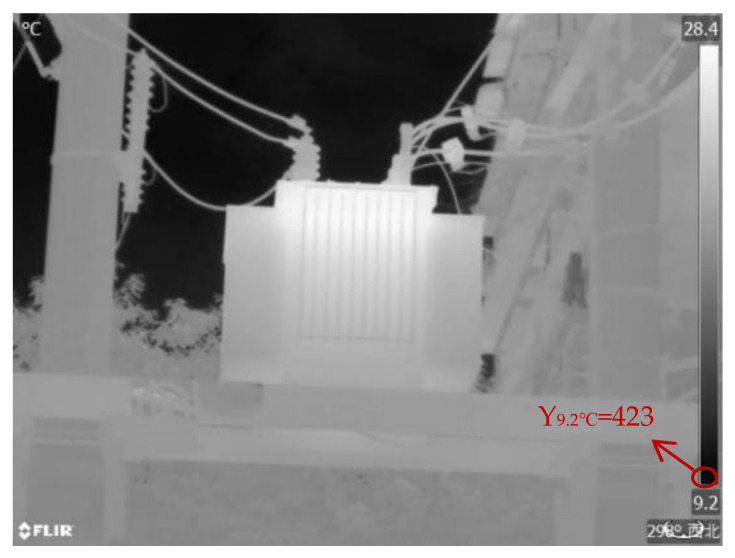
Grayscale chart.

**Figure 11 sensors-24-01126-f011:**
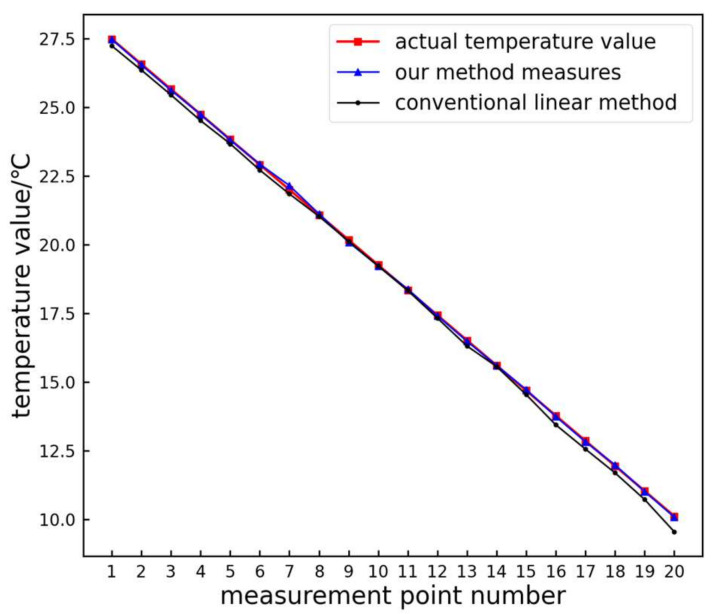
Measurement point temperature measurement results.

**Figure 12 sensors-24-01126-f012:**
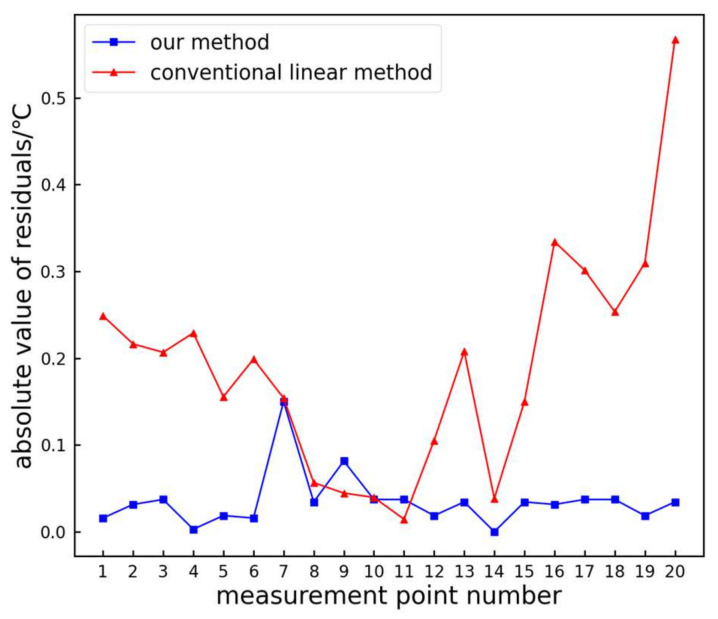
Comparison of absolute residual values.

**Figure 13 sensors-24-01126-f013:**
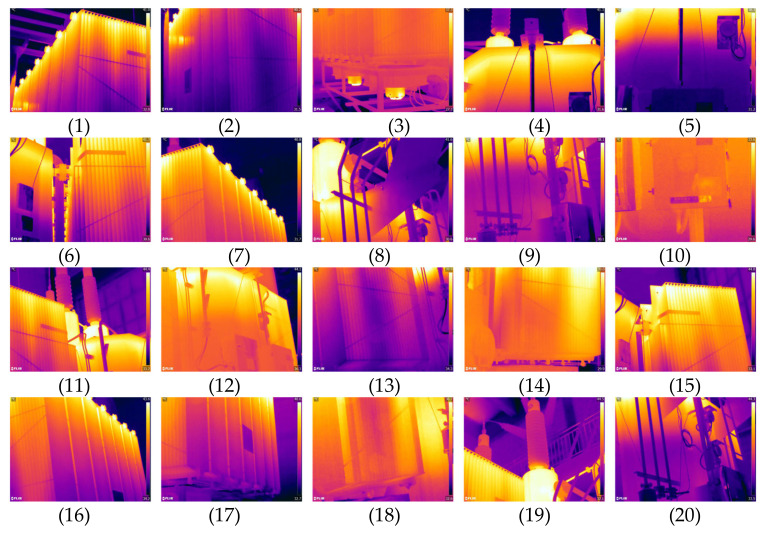
Experimental infrared images. (1)–(20) represents the experimental image number.

**Figure 14 sensors-24-01126-f014:**
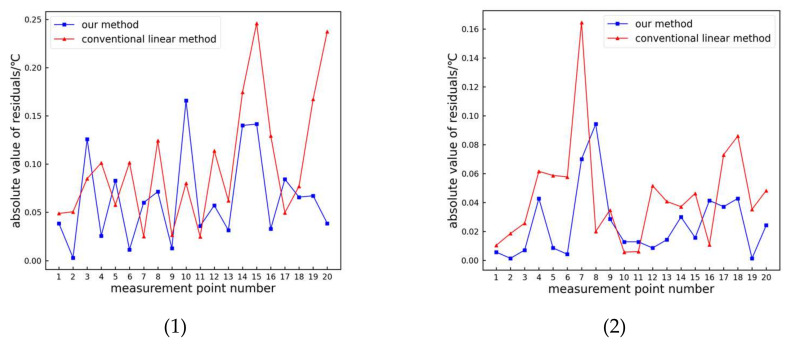
Comparison of the absolute residual values. (1)–(20) represents the experimental result number corresponding to each image in Figure 13.

**Figure 15 sensors-24-01126-f015:**
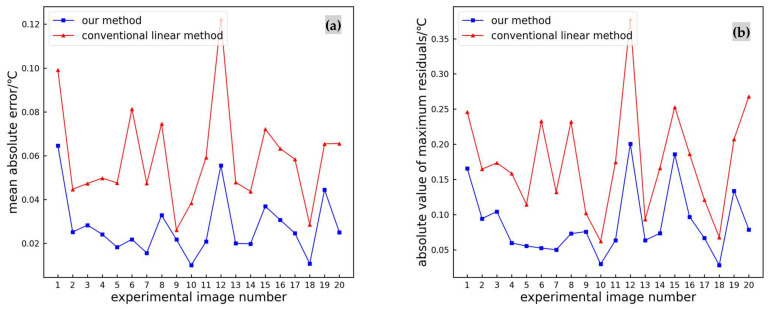
Error per infrared image: (**a**) mean absolute error; (**b**) absolute value of maximum residuals.

**Figure 16 sensors-24-01126-f016:**
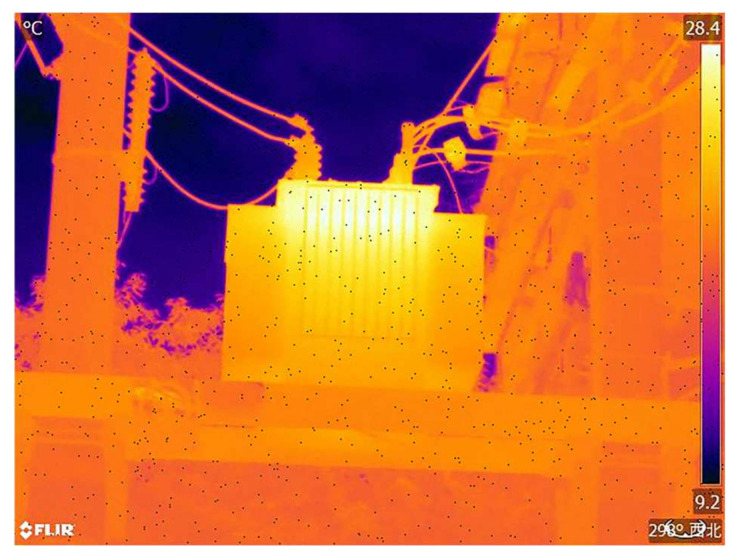
Infrared image after adding noise.

**Figure 17 sensors-24-01126-f017:**
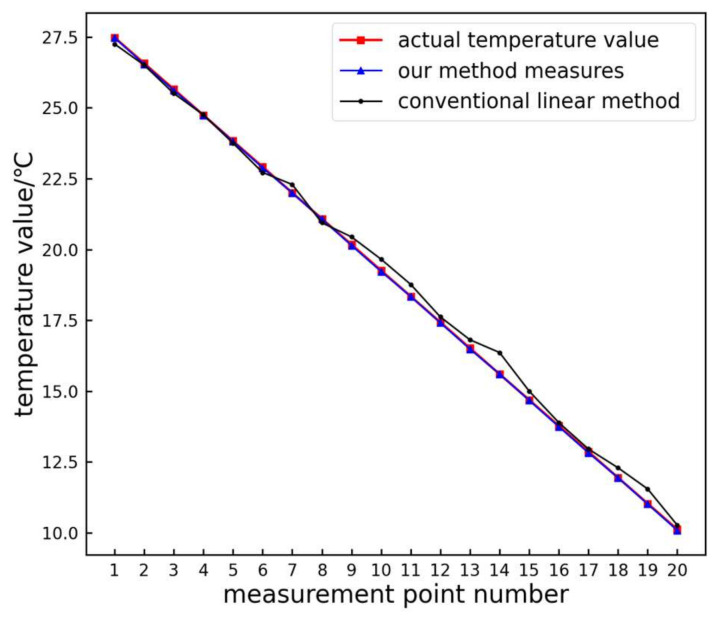
Adding noise measurement point temperature measurement results.

**Figure 18 sensors-24-01126-f018:**
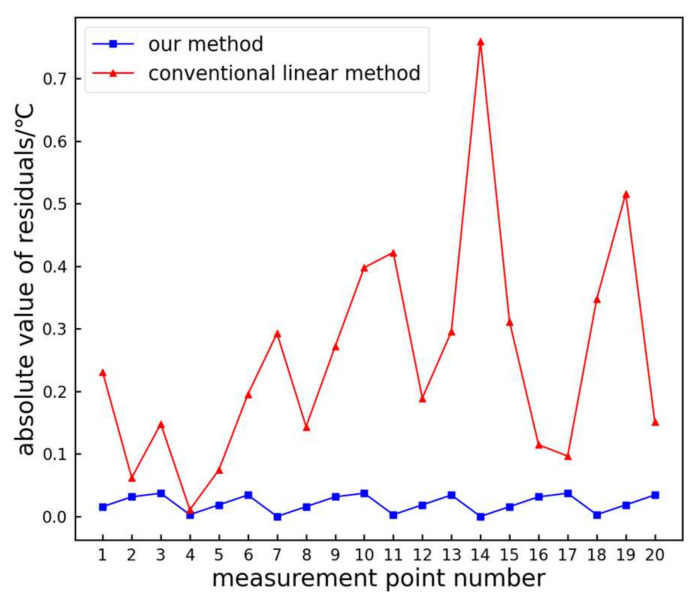
Comparison of the absolute values of residuals.

**Figure 19 sensors-24-01126-f019:**
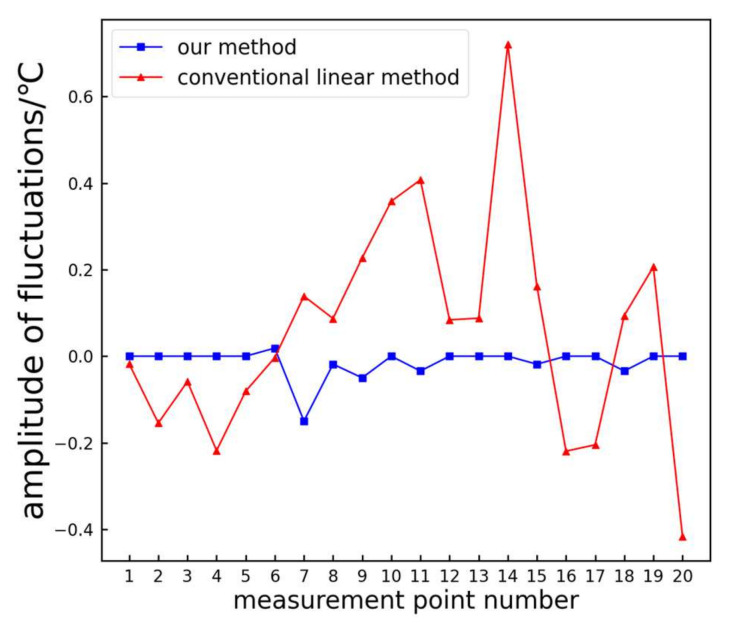
Temperature value fluctuation comparison chart.

**Table 1 sensors-24-01126-t001:** Test results.

Compared Categories	Image	GWO–OTSU	SSA–OTSU	GA–OTSU	WOA–OTSU	OTSU
Threshold	(a)	90	90	92	90	90
	(b)	93	93	93	93	93
	(c)	112	112	111	113	112
	(d)	96	96	94	94	96
PSNR	(a)	27.91	27.91	27.92	27.91	27.91
	(b)	27.40	27.40	27.40	27.40	27.40
	(c)	28.41	28.41	28.34	28.40	28.41
	(d)	28.01	28.01	28.00	28.00	28.01
SSIM	(a)	0.5030	0.5030	0.4999	0.5030	0.5030
	(b)	0.3624	0.3624	0.3624	0.3624	0.3624
	(c)	0.4723	0.4723	0.4787	0.4665	0.4723
	(d)	0.2390	0.2390	0.2414	0.2414	0.2390
Elapsed time/ms	(a)	21.87	43.76	42.29	38.68	147.82
	(b)	21.77	43.86	39.83	38.63	134.38
	(c)	22.04	42.94	40.62	38.26	133.76
	(d)	21.98	44.24	39.76	38.13	131.56
Average running time/ms		21.915	43.7	40.625	38.425	136.88
Average uptime lift rate/%		83.99	68.07	70.32	71.93	

**Table 2 sensors-24-01126-t002:** Test results.

Compared Categories	Image	GWO–OTSU	SSA–OTSU	GA–OTSU	WOA–OTSU	OTSU
Threshold	(1)	50	50	50	51	50
	(2)	84	84	83	85	84
	(3)	28	28	28	27	28
	(4)	63	63	63	63	63
	(5)	54	54	54	54	54
	(6)	77	77	78	78	77
	(7)	41	41	39	42	41
	(8)	33	33	34	33	33
	(9)	43	43	43	42	43
	(10)	37	37	36	38	37
	(11)	67	67	64	69	67
	(12)	76	76	77	74	76
	(13)	35	35	32	34	35
	(14)	84	84	84	82	84
	(15)	16	16	19	16	16
	(16)	31	31	33	30	31
	(17)	62	62	59	63	62
	(18)	45	45	45	46	45
	(19)	66	66	66	69	66
	(20)	60	60	63	58	60
Elapsed time/ms	(1)	22.66	43.96	40.63	37.08	131.47
	(2)	22.15	43.79	37.12	40.07	131.16
	(3)	22.29	44.76	39.57	39.59	136.79
	(4)	22.87	45.83	40.96	36.65	133.47
	(5)	21.85	44.66	40.08	38.09	139.70
	(6)	19.99	42.37	36.50	42.55	131.48
	(7)	21.01	42.67	34.42	38.40	133.83
	(8)	20.29	42.53	39.08	37.19	132.44
	(9)	20.19	41.64	33.96	34.69	132.31
	(10)	21.19	42.02	38.04	37.92	134.32
	(11)	20.79	40.70	35.23	35.58	133.84
	(12)	20.22	44.27	34.52	37.59	133.85
	(13)	21.45	41.37	32.92	38.22	139.19
	(14)	20.87	42.42	39.84	37.78	131.57
	(15)	20.98	42.48	42.56	35.42	131.56
	(16)	21.97	42.50	38.28	35.77	131.17
	(17)	20.51	42.46	38.51	37.09	130.29
	(18)	20.77	42.35	42.08	35.99	132.69
	(19)	20.50	43.87	39.10	37.96	132.79
	(20)	21.83	43.67	40.77	37.65	135.10
Average running time/ms		21.219	43.016	38.2085	37.564	133.451
Average uptime lift rate/%		84.10	67.77	71.37	71.85	

## Data Availability

Data are contained within this article.

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
