# Peer review of "Optimized OTSU Segmentation Algorithm-Based Temperature Feature Extraction Method for Infrared Images of Electrical Equipment"

_sensors, 2024, doi:10.3390/s24041126_

Round 1

Reviewer 1 Report

Comments and Suggestions for Authors

In this article, the author introduces a GWO-OTSU infrared image segmentation method and an algorithm for extracting temperature values from infrared images of power equipment using KNN. The author designs experiments to verify the algorithm, which has reference significance. I have the following suggestions for this article:

 1. In abstract, it was mentioned that “increases the threshold calculation time by more than 83.88%”, but in the entire text, only 83.88% appeared once in the abstract (line 21), and other experimental results showed a decrease of 83.99% (line 324, line 513).

 2. The statement in line 198 seems difficult to understand, is there some missing information?

 3. The article believes that “device segmentation” is a relatively important part. In the part of related works, the author cited several references, and focus on traditional image segmentation methods. Why not introduce more popular deep learning methods of image segmentation?

 4. Line 280 says “In this paper, Lena image, infrared images of operating electrical equipment (Fig.6), and cropped portions of insulators, transformers, and casings from these infrared images were selected as experimental images.”

But I didn't see Lena image and experiments in the paper. Furthermore, how does the GWO-OTSU algorithm proposed in this article perform on public a datasets?

 5. Line 408 “The actual temperature values of all points within the rectangular box are as follows: 28.383℃, 26.652℃, and 27.826℃, respectively.” Which is different from other experiments. These results are confused, and the author should explain it clear. How to obtain them? Is it obtained based on formula (11) or “accompanying infrared image analysis software”?

 6. Reference 21 seems to point to an invalid link.

 7. In Figures 12, 14, 15, 18, and 19, the author presents the relationship between the absolute residual values, why not give residuals directly?

  8. The section “2.2 GWO optimized OTSU segmentation algorithm” should be more specific, as it now appears to be only introducing GWO. In addition, the title of section 2.2 is duplicated with the title of chapter 2.

Comments on the Quality of English Language

Extensive editing of English language required

Reviewer 2 Report

Comments and Suggestions for Authors

The paper ''A Method of Temperature Feature Extraction on Infrared Images of Electrical Equipment Based on Optimized Otsu Segmentation Algorithm'' is well conceived and methodologically correct.

In order to solve the problem of limited research related to the non-linear relationship between grayscale and temperature and the problem of improving the accuracy of extraction when extracting temperature information using the relationship between grayscale and temperature, the authors proposed a temperature feature extraction method for infrared images of electrical equipment based on an optimized Otsu algorithm. The Gray Wolf Optimization (GWO) algorithm was used to optimize the threshold determination process in the traditional Otsu segmentation method, resulting in an improved Otsu segmentation algorithm based on GWO. That enhanced the segmentation speed and separates the target device regions in the infrared image (experimental results show that the optimized segmentation method, compared to the non-optimized method, increases the threshold calculation time by more than 83.88% while maintaining similar segmentation results). Based on the obtained results, the authors were proposed a temperature extraction method for infrared images using the K-Nearest Neighbors (KNN) algorithm to improve the temperature value extraction accuracy. Experimental results demonstrate that compared to traditional linear methods, this method achieves a 73.68% improvement in the maximum residual absolute value of the extracted temperature values and a 78.95% improvement in the average residual absolute value.

In addition to the introduction and conclusion, the paper consists of three more sections. In introduction the authors explained the research problem and analyzed literature. In order to improve the segmentation speed, the GWO optimization algorithm is utilized in Section 2 to optimize the threshold determination process in OTSU. The optimized threshold obtained in section 2 is then used for infrared image segmentation which is described in Section 3. The optimized threshold obtained in Section 2 was then used for the infrared image segmentation described in Section 3. The validation of the proposed method is performed in Section 4.

In conclusion, the authors briefly described the contribution of the research.

There are 19 figures and two tables in the paper, which serve to better illustrate the research. The research design is appropriate. The cited references are relevant to the research. There are some grammatical errors in the paper. The directions of future research are missing.

Remarks:

In conclusion, it is necessary to specify the directions of future research.

It would be nice if the authors check the English language and correct the grammatical errors in this paper.

The paper can be published after minor revision.

Reviewer 3 Report

Comments and Suggestions for Authors

Comments to the authors:

1-      In Figures 1 and 3, the text of the flowchart should be bigger in order to be more comfortable for the readers.

2-      Page 6, line 229, what are the authors mean by “gray value”?

3-      Page 7, line 246, why max, min and average temperature are selected to extract?

4-      Page 8, line 296, the authors need to explain in detail how the optimal thresholds are obtained?

Round 2

Reviewer 1 Report

Comments and Suggestions for Authors

 Minor editing of English language required

Comments on the Quality of English Language

Minor editing of English language required

Reviewer 3 Report

Comments and Suggestions for Authors

My remarks have been addressed properly. I think, it is OK now and can be published.